# Stem Cell Strategies in Promoting Neuronal Regeneration after Spinal Cord Injury: A Systematic Review

**DOI:** 10.3390/ijms232112996

**Published:** 2022-10-27

**Authors:** Lapo Bonosi, Manikon Poullay Silven, Antonio Alessandro Biancardino, Andrea Sciortino, Giuseppe Roberto Giammalva, Alba Scerrati, Carmelo Lucio Sturiale, Alessio Albanese, Silvana Tumbiolo, Massimiliano Visocchi, Domenico Gerardo Iacopino, Rosario Maugeri

**Affiliations:** 1Neurosurgical Clinic, AOUP “Paolo Giaccone”, Post Graduate Residency Program in Neurologic Surgery, Department of Biomedicine Neurosciences and Advanced Diagnostics, School of Medicine, University of Palermo, 90127 Palermo, Italy; 2Department of Traslational Medicine, University of Ferrara, 44121 Ferrara, Italy; 3Department of Neurosurgery, Sant’Anna University Hospital of Ferrara, 44124 Ferrara, Italy; 4Department of Neurosurgery, Fondazione Policlinico Universitario “A. Gemelli” IRCCS, Università Cattolica del Sacro Cuore, 00100 Rome, Italy; 5Division of Neurosurgery, Villa Sofia Hospital, 90146 Palermo, Italy

**Keywords:** bioengineering, nanotechnology, scaffolds, neuronal regeneration, spinal cord injury, stem cell therapy

## Abstract

Spinal cord injury (SCI) is a devastating condition with a significant medical and socioeconomic impact. To date, no effective treatment is available that can enable neuronal regeneration and recovery of function at the damaged level. This is thought to be due to scar formation, axonal degeneration and a strong inflammatory response inducing a loss of neurons followed by a cascade of events that leads to further spinal cord damage. Many experimental studies demonstrate the therapeutic effect of stem cells in SCI due to their ability to differentiate into neuronal cells and release neurotrophic factors. Therefore, it appears to be a valid strategy to use in the field of regenerative medicine. This review aims to provide an up-to-date summary of the current research status, challenges, and future directions for stem cell therapy in SCI models, providing an overview of this constantly evolving and promising field.

## 1. Introduction

The worldwide incidence of spinal cord injury (SCI) ranges from 10.4 to 83 cases per million persons per year [1,2]. SCI results in significant morbidity and mortality and is associated with lifetime health care costs, representing a considerable burden to the community and society. SCI affects men more often than women, with 78% of new cases being male [3]. The average age of patients at the time of injury is approximately 43 years. Still, there is a bimodal distribution with a peak in adolescents and young adults and a second peak in the elderly population (>65 years) [4].

SCI can be divided into traumatic and non-traumatic spinal cord injury. Traumatic SCI is more common and typically caused by external physical impacts, such as falls or traffic accidents. Otherwise, tumor compression, vascular ischemia, or congenital disease are common causes of non-traumatic SCI [5,6].

Pathophysiology of SCI includes cell death, axonal collapse and demyelination, glial scar formation, inflammation, and other pathological defects [7]. Following contusion injury, the equilibrium of the spinal cord microenvironment is disrupted: downregulation of beneficial factors and an upregulation of harmful ones [8]. These imbalances impair regeneration and functional recovery [9]. Spinal cord damage consists of two distinct phases: primary or direct injury and secondary or indirect injury. The first is a result of blunt trauma, resulting in shearing and laceration of spinal cord fibers owing to an acceleration-deceleration mechanism. The impact can result in a contused spinal cord and, rarely, complete transection [10]. The overall primary phase consists of axonal injury, disruption of blood vessels, and disruption of cellular membranes.

The latter is the physiologic response to the initial trauma causing inflammation, ischemia, vascular dysfunction, free-radical formation, impaired neuronal hemostasis, and apoptosis or necrosis. Indirect injury comprises neurons and glia death with cell necrosis. This results in increased inflammation, edema, and hemorrhage with further progression of axonal damage and cell necrosis. The subacute phase is characterized by a phagocytic response to clear cell debris and initiate early axonal growth. During this phase, the damaged astrocytes die, while the astrocytes on the periphery of the damaged tissue proliferate. A harmful consequence of astrocytic proliferation is scar formation, which prevents axonal regeneration because the scar acts as a physical and chemical barrier. In the final stages, scar evolution and syrinx formation occur with myelomalacia and cystic cavitations (Figure 1). Potential therapeutic interventions target these primary and secondary stages to optimize spinal cord recovery and regeneration [11,12,13].

However, numerous inflammatory molecules, a particular microenvironment, and bioumoral factors hinder neuronal regeneration and recovery of function in the Central Nervous System (CNS) [14]. Conversely, in the Peripheral Nervous System (PNS), axonal remyelination begins immediately after injury and acts more efficiently than in the CNS. The key difference between CNS and PNS regeneration is the type of glial cells. Schwann cells and oligodendrocyte cells are the main glial cells responsible for neuronal support and myelination in the PNS and CNS, respectively [15]. Specifically, while dysregulation of CNS oligodendrocytes and astrocytes following injury is partly responsible for the inhibition of repair mechanisms through the release of inhibitory molecules, in the PNS, Schwann cells and Satellite cells induce neuronal regeneration through the release of neurotransmitters, growth factors, and specific intracellular signaling molecules [16,17,18].

Nevertheless, recent studies have shown that even at the CNS level, glial cells and some immunity cells play a dual role depending on their phenotype. Particularly, microglia can trigger the activation of genes responsible for the dysregulated microenvironment within the lesion site. It has been demonstrated that there is a time-dependent transformation of reactive microglia and astrocytes into their neuroprotective phenotypes (M2a, M2c and A2) which are crucial for post-SCI spontaneous locomotor recovery [19]. They can change their phenotype and functions in response to injury-related factors [20]. For instance, myelinated debris at the lesion site switches macrophages from the M2 to the M1-like pro-inflammatory phenotype. Additionally, myelinated debris activates the ATP-binding cassette transporter A1 (ABCA1) for cholesterol efflux. This dysregulation in the homeostatic mechanism leads to the development of foamy macrophages and lipid plaques at the injury site, inducing a pro-inflammatory environment associated with increased neurotoxicity and impaired wound healing [21]. Furthermore, damage-associated molecular patterns (DAMPs) released by necrotic cells after injury worsen inflammation. High mobility group box 1 (HMGB1), an identified DAMP, is a ubiquitously expressed DNA binding protein. It has been shown that reactive astrocytes could undergo necroptosis, releasing HMGB1 after SCI in mice models [22]. Furthermore, HMGB1 appears to induce pro-inflammatory microglia transformation via the RAGE-nuclear factor-kappa B (NF-kB) pathway. Notably, inhibition of HMGB1 or RAGE significantly reduces neuronal loss and demyelination and improves functional recovery after SCI [23]. On the other hand, tissue repair after SCI requires mobilizing immune and glial cells to form a protective barrier that seals the wound and facilitates debris cleaning, inflammatory containment, and matrix compaction. This process, termed corralling, involves phagocytic immune cells and microglia. Corralling is an important step to mitigate secondary tissue injury fueled by inflammatory cytokines, proteases and free-radicals released from the lesion core, thus promoting early functional axonal recovery [24,25].

A temporally distinct gene signature in microglia and lesion-activated macrophages (IAMs) has been discovered that engages the axon guide pathways [26]. Plexin-B2 is an axon guidance receptor widely expressed in the developing brain and involved in SCI recovery [27]. It is upregulated in IAMs and seems to promote motor and sensory recovery after SCI. Corralling requires Plexin-B2 in both microglia and macrophages. Mechanically, Plexin-B2 promotes microglia motility, moves IAMs away from colliding cells and facilitates matrix compaction. Therefore, Plexin-B2 is an important link that integrates the biochemical signals and physical interactions of IAMs with the wound microenvironment during wound healing [28,29]. Deletion of Plexin-B2 in myeloid cells leads to diffuse tissue damage, inflammatory spillovers, and hinders axon regeneration.

The need to find effective and safe regenerative strategies that promote CNS tissue repair becomes clear in this context. Over the last decades, different regenerative strategies have been suggested for promoting brain or spinal cord repair, including direct cell transplantation, direct growth factor injections, and tissue engineering strategies based on the combination of biomaterial, stem cells, and growth factors [30]. Despite the continuous crosstalk between bioengineering and medicine, there are currently no effective regenerative treatments. In this scenario, stem cell-based therapy is a promising treatment for SCI due to its multiple targets and reactivity benefits. Stem cell-based therapies in SCI have different mechanisms in functional recovery, such as immunomodulation, cell replacement nutrition, and scaffold support.

The present review focuses on SCI stem cell therapy, including bone marrow mesenchymal stem cells, umbilical mesenchymal stem cells, adipose-derived mesenchymal stem cells, neural stem cells, and neural progenitor and embryonic stem cells. Each cell type targets certain features of SCI pathology and shows therapeutic effects via cell replacement, nutritional support, scaffolds, and immunomodulation mechanisms. This review aims to perform an up-to-date summary of the current research status, challenges, and future directions for stem cell therapy in SCI, providing an overview of this constantly evolving field.

## 2. Methods

### 2.1. Search of the Literature

A systematic literature review was conducted following the PRISMA (Preferred Reporting Items for Systematic Reviews and Meta-Analyses) guidelines [31]. A time limit was chosen because the research in this area is rapidly evolving. We aimed to get a picture of the state-of-the-art regarding stem cell therapy, its potential applicability in SCI settings and current limitations.

The following MeSH (Medical Subject Headings) terms were used: “Bioengineering”, “Nanotechnology”, “Biotechnology”, “Neuronal Regeneration”, “Spinal Cord Injury”, “SCI”, “Stem Cell Therapy”, “Spine Surgery”, combined using Boolean operators “AND” and “OR”. Duplicate articles were eliminated using Microsoft Excel 16.37 (Redmond, WA, USA).

### 2.2. Study Selection

A manual search was conducted to identify eventual studies and other reference sections. Four reviewers then independently screened the titles, abstracts, and full manuscripts of the articles first included in the database; then, the results were analyzed and combined.

Our research was initially focused on all methods of neuronal regeneration, affecting both the CNS and PNS. We later decided to direct our research to spinal cord injuries, given the great importance of this within the neurosurgical field. No ethical approval was required for this review.

### 2.3. Eligibility Criteria

The articles were selected according to the following criteria:Full article in EnglishExperimental studies conducted in vivoStudies investigating stem cell related neuronal regeneration techniques after SCI

Exclusion Criteria:Articles not in EnglishLiterature reviews, systematic reviews, meta-analysisArticles published before 2008Studies conducting only in vitro experimentsStudies focusing on PNS injuryStudies focusing on techniques not related to stem cells

## 3. Results

### 3.1. Study Selection

The initial search through the PubMed^®^ database and references section screening yielded 1147 results. After duplicate removal (n = 178), the initial database was composed of 970 articles. The first selection was made by title, rejecting 447 articles. In total, 523 abstracts were screened according to the selection criteria. Of those, 422 articles were excluded because the abstract was considered noninherent to the purpose of our review, and the other 13 because we could not retrieve the full text. Eighty-eight articles were deemed eligible for full-text review. We then excluded 61 studies because they did not match our inclusion criteria. Finally, we included 27 articles in this review (Figure 2).

### 3.2. Study Characteristics

All included articles are experimental studies conducted on in vivo models of rats. Only one study was conducted on two exemplars of Macacus Rhesus. Each work examines neuronal regeneration and functional recovery on models of SCI. The techniques used involve implanting biomaterials, such as scaffolds enriched with stem cells, injection of stem cells at the lesion site, and peripheral administration of stem cells. Of the 27 studies selected: 15 (55.5%) focused their attention on scaffold-related techniques combined with a different type of stem cell or scaffold structure. All of them were prospective experimental studies (100%); 26 used in vivo models of rats (96.3%), and only one study was conducted on two exemplars of Macacus Rhesus (3.7%); 7 studies used injection of stem cells at the lesion site (25.9%), and 1 evaluated a systemic infusion of stem cells (3.7%). One study evaluated the administration of melatonin associated with physical exercise (3.7%). (Table 1).

### 3.3. Study Synthesis

All included articles are experimental studies conducted on in vivo models, examining neuronal regeneration in animal models of SCI. The techniques used involve implanting biomaterials, such as scaffolds enriched with stem cells, injection of stem cells at the lesion site, and peripheral administration of stem cells, and in one case, the administration of melatonin associated with physical exercise. Outcome analysis was then evaluated by immunohistochemical, histological, and microscopic investigations and in in vivo models by observation of the functional motor recovery of the animals examined. What emerges from all papers is the close relationship between bioengineering, biotechnology, and medicine, reflecting the great importance and role of an interdisciplinary approach in this complex research field.

## 4. Discussion

Stem cell therapy is a promising treatment for SCI due to its multiple targets and reactivity benefits. Various stem cell lines can be used, each associated with specific advantages and disadvantages (Table 2).

### 4.1. Stem Cells and Scaffolds Strategies

Neural stem cells (NSCs) are pluripotent cells capable of differentiating into specific neuronal or glial cells, enhancing remyelination, and providing nutritional support. Moreover, NSCs are suitable for basic spinal cord tissue engineering research as they can constitutively secrete brain-derived neurotropic factors and differentiate into neurons [59]. These features make them suitable for cell transplantation therapy in SCI, but whether they effectively improve the patients’ functions remains controversial [60,61].

In addition to the source and type of stem cell used, a key role in the process of tissue regeneration is played by the scaffolds, matrices, and enzymes that can be associated together with stem cells to promote their rooting, counteract any inhibitory agents at the level of the damaged site, or limit therapy-related adverse effects. For example, a promising technique is exosome-based cellular therapy. It is a tissue engineering method characterized by several advantages, such as zeroing the risk of immunological rejection, tumorigenicity, and vascular occlusion. According to Zhong D et al., NSCs-Exosomes exhibited a proangiogenic effect on SCMECs (spinal cord microvascular endothelial cells) by transferring Vascular Endothelial Growth Factor A (VEGF-A) and thus promoting microvascular regeneration and tissue healing. In vitro studies in SCI mice have documented an increase in local microvascular density, spinal cord cavity shrinkage and recovery of motor function [62].

As mentioned above, the essential elements of tissue bioengineering include seed cells, scaffolds, and growth factors. Several scaffolds have been created in recent years, but no ideal material to simulate the ultrastructure of the spinal cord was found. Wan et al. [33] constructed a self-polymerized scaffold for spinal cord tissue engineering using a peptide self-polymerization nano gel. The gel has been observed to have fine viscoelasticity simulating the biomechanical properties of the spinal cord, thus promoting an optimal microenvironment for axonal regeneration [63,64,65]. Conversely, Ciciriello et al. [32] compared the NSC of adult mice vs. E14 (embryonic on the 14th day) mice NSC on a multichannel PLG bridge to improve the number and extent of myelination for axons growing in and through the lesion. Their results reveal that cell survival correlated with a commensurate increase in the density of axons and myelin for mice receiving E14 transplants compared to age-matched adult transplants. Furthermore, it has been demonstrated that PLG bridges constitute a promising biomaterial platform with limited scar formation and guidance of axon elongation through the injury [66,67,68,69]. Yuan and colleagues [70] focused on the role of adipose-derived stem cells (ADSCs) as a potential regenerative source for SCI therapy. They have developed a cell-adaptable neurogenic (CaNeu) hydrogel to deliver ADSCs, thus promoting neuronal regeneration after tSCI. CaNeu hydrogel loaded with ADSC significantly increased the M2 macrophage population by minimizing the harmful effects of inflammation at an early stage and promoting nerve fiber regeneration. To date, several preclinical studies have demonstrated the potential ability of ADSCs to repair SCI in a paracrine manner. A clinical trial investigating the use of autologous ADSCs for treating patients with cervical SCI is currently underway at the Mayo Clinic [71,72,73]. Farrag et al. [40] investigated the effects of the subcutaneous maturation of adult-derived neural stem cells (aNSCs) seeded into biomaterial constructs, revealing that the hydrogels supported aNSC survival and differentiation for up to 4 weeks in the subcutaneous environment. This study lays the groundwork for a bioengineering approach that can form region-specific neuroepithelium. Histological analysis revealed that aNSCs clustered cell formations throughout the constructs that highly resemble neural rosettes, a hallmark of CNS development, which expressed neuroepithelial markers (PAX6 and Sox1) along with Nestin, the NSC marker [74,75]. Other studies have focused on using various types of stem cells associated with different types of matrices or scaffolds, intending to enhance neuronal regeneration, limit scar formation, or inhibit the inflammatory microenvironment resulting from SCI [44,45,47,48]. The results are encouraging, but further investigations are essential to assess their clinical applicability (Figure 3).

### 4.2. Combinatory Strategies

New combinatorial strategies for the treatment of SCI have been investigated in recent years, showing enormous potentiality, and obtaining encouraging results in preclinical models. In this context, Yang and colleagues [34] exploited a combined strategy to stimulate axon regeneration and functional recovery after complete resectioning of the thoracic spinal cord in adult rats. They used a multichannel poly lactic-co-glycolic acid (PLGA) scaffold seeded with activated Schwann cells (ASCs) and mesenchymal stem cells (MSCs) to create a bridge between the rostral and caudal abutments [76,77,78]. ASCs enhance axonal regeneration and functional recovery in SCI models by expressing neurotrophic factors and cell adhesion molecules [79,80,81]. On the other hand, PLGA scaffolds prevented cyst formation and supported tissue regeneration at the injury/graft site. Moreover, after in vitro co-culture, it has been shown that ASCs promote MSC differentiation into neuron-like cells expressing immature and mature neuronal markers, such as β-tubulin three and MAP2 [82,83]. Remarkably, a partial restoration of the latencies and amplitudes of sensory evoked potential (SEP) and motor evoked potential (MEP) was observed, indicating that the number of regenerated axons and the nerve conduction velocity could be recovered through this combinatorial strategy. Likewise, the high level of neuronal differentiation obtained can be attributable to neurotrophic factors, including glial cell line-derived neurotrophic factor (GDNF) mRNA and brain-derived neurotrophic factor (BDNF) mRNA, expressed in activated Schwann cells [79,84,85]. In the study by Zhou et al. [36], human umbilical cord blood-induced pluripotent stem cells (hUCB-iPSCs) were induced in NSC as “seed cells” and were co-cultured with ASC [86]. It has been demonstrated that hUCB-iPSCs reduce the lesion cavity’s volume and improve tSCI rat models’ locomotor recovery. In addition, they emphasized that the degree of spinal cord recovery and remodeling may be closely related to nerve growth factors and glial cell-derived neurotrophic factor. For instance, GDNF is one of the most relevant neurotrophic factors capable of influencing the degree of recovery and remodeling of the spinal cord by influencing the behavior of glial cells, in addition to the regeneration of axons [87,88]. Finally, Fan et al. [49] proposed the combination of a 3D gelatin methacrylate (GelMA) hydrogel with iPSC-derived NSCs (iNSCs) to promote regeneration after SCI. They discovered that GelMA/iNSC implants significantly promote functional recovery, reduce cavity areas, and inhibit collagen deposition. GelMA/iNSC implantation showed striking therapeutic effects of inhibiting GFAP-positive cells and glial scar formation while simultaneously promoting axonal regeneration.

### 4.3. Usefulness of Enzyme Strategy and Role of Neurotrophic Factors

A crucial aspect that needs to be taken into account in neuronal regeneration strategies is that related to the inflammatory microenvironment and the enzymic kit that accompanies neuronal damage in SCI. Führmann et al., studied the role of chondroitin sulfate proteoglycans (CSPGs) expression in SCI animal models treated with and without chondroitinase ABC (chABC) [35]. They have observed more significant neuronal differentiation when neuroepithelial stem cells (NESCs) were supplied with chABC, suggesting that CSPGs play a role in inhibiting the differentiation and survival of NESCs. Furthermore, the delivery of chABC attenuates the inflammatory response, beneficial to neuron survival and host tissue regeneration [89,90]. Despite the increased neuronal differentiation, no behavioral recovery was observed with the combined treatment. This is probably due to the poor survival of immunocompetent human cells in animals, even in the presence of immunosuppression [91,92]. Furthermore, few cells demonstrated mature neuronal markers, indicating that longer survival times may be required to have a greater effect on motor recovery. Nori and colleagues investigated the role of chABC delivery for the treatment of SCI in an immunodeficient rat model, finding an increased long-term survival of NPCs around the lesion epicenter, a meaningful oligodendrocyte differentiation, higher remyelination rate of the spared axons, and enhanced synaptic connectivity with anterior horn cells associated with higher neurobehavioral recovery [42]. Interestingly, previous studies have indicated that the limited distribution of grafted NPCs due to the glial scar results in a lack of functional recovery in SCI [93].

Lastly, many efforts have been directed on the utility and role of neurotrophic factors in the process of neuronal regeneration and the study of molecular pathways underlying both damage and reparative mechanisms in SCI. For instance, Hwang et al., aimed to evaluate whether glial cell line-derived neurotrophic factor (GDNF) would augment the therapeutic effects of Neural stem/progenitor cell (NSPCs) SCI treatment. To do so, GDNF-encoding, or mock adenoviral vector-transduced human NSPCs (GDNF-or Mock-hNSPCs) were transplanted into the injured thoracic spinal cords of rats seven days after SCI. The results showed significantly reduced lesion volume and glial scar formation, promoted neurite outgrowth, axonal regeneration and myelination, increased Schwann cell migration that contributed to the myelin repair, and improved locomotor recovery [39]. Instead, Yan et al. [37], starting from the concept that myelin stimulates axonal regeneration from mammalian neuronal progenitor cells (NPCs), have hypothesized that myelin-associated proteins may contribute to axonal regeneration from NPCs. In their work, they discovered that myelin basic protein (Mbp) supports axonal regeneration from mammalian NPCs through the novel Mbp/L1cam/Pparγ signaling pathway, suggesting that bioengineered, NPCs-based interventions can promote axonal regeneration and functional recovery post-SCI. The stimulatory effects of Mbp on neurite growth in NPCs are mediated by the production of L1-70, which fosters neuritogenesis, axonal regeneration, and post-SCI functional recovery [38,50,51,52]. Using genetic overexpression of trophic molecules in cell populations has been a promising strategy for developing cell replacement therapies for spinal cord injury (SCI). Notably, it has been shown how the overexpression of SDF-1 by MSCs can enhance the migration of NSCs in vitro. Although only modest functional improvements were observed following transplantation of SDF-1-MSCs, a significant reduction in cavitation surrounding the lesion occurred [41].

### 4.4. Limits of Stem Cell-Based Therapy

Although stem cells represent an attractive therapeutic strategy, they still have several limitations [53]. Some risks include tumorigenesis, immunological complications, allodynia, or complications associated with an unexpected change in phenotype of the transplanted cells (i.e., dedifferentiation or excess proliferation) [54]. Tumor formation is a significant concern for transplant strategies involving embryonic stem cells; however, this risk decreases as the cells become more highly differentiated. Futhermore, tumor formation is of particular concern because the cells implanted in the spinal cord would be difficult or impossible to remove [55]. Another important limitation is immune rejection of the transplant. The host’s immune system could destroy the implanted cells. Human Leukocyte Antigen (HLA) matching of the stem cell transplant to the host is one method to avoid immune rejection. In other cases, life-long immunosuppression may be necessary to prevent rejection of the transplanted cells, with possible ADEs (adverse drug-related events) [56]. Stem cell-based strategies to treat SCI can cause allodynia. In some studies, neural stem cells were transplanted into the low-thoracic spinal cord of rats 1 week after injury. Functional recovery was noted in the affected hind limbs, but abnormal, painful sensitivity developed in the forepaws [57]. Differentiation to undesirable cell types is a risk inherent to all multipotent cells. In addition to evaluating the phenotype, preclinical studies should also assess the proliferation and migration of the transplanted cells. Uncontrolled proliferation and migration are obviously undesirable; spreading beyond the implant site would increase the risk of adverse events such as cerebrospinal fluid occlusion or emboli causing stroke. Cell migration (bio-distribution) from the implant site can be evaluated in preclinical testing using reverse transcription polymerase chain reaction (RT-PCR) for specific markers on isolated organs following transplantation. Inappropriate cellular differentiation must also be defined in preclinical testing [54].

Finally, other limitations exist for using MSCs as a therapeutic tool for SCI. One of the most important issues is the selection of the best cell transplantation routes. A disadvantage of MSC infusion could be cell trapping in other organs, as well as the risk of immune reactions, the low neural differentiation rate, and low survival rate. Given the potential of MSCs in SCI, many researchers have searched novel strategies to promote MSC engrafts. In this regard, combining MSCs with scaffolds is a promising strategy to promote MSC survival, proliferation, and differentiation [58,94]. However, most of the clinical trials are still ongoing; therefore, data regarding safety, efficacy and side effects are not yet available. Much information will be available at the end of the year 2022 when the data on clinical trials currently underway will be published [53].

### 4.5. Alternative Methods to Promote Repair and Regeneration of Nerve Tissue

Apart from stem cell-based therapy, other methods are being studied to promote the repair and regeneration of damaged nerve tissue in SCI [95,96]. One option is autografting neural tissue transplantation combined, or not, with functional biomaterials [97]. In vivo studies in rat models have shown encouraging prospects about the possibility of using this type of treatment in the future to treat SCI [98]. Another option is allograft nerve cell transplantation. A study focusing on the peripheral nervous system has shown signs of regeneration and clinical improvement for patients undergoing this treatment [99]. However, this strategy is burdened by an increased risk of rejection, compared with the first option, and further studies are needed to test its safety and applicability. Another field that researchers have been devoted to is stimulation by electrical impulses. As a matter of fact, it is bolstered that electrical stimuli promote nerve regeneration and guide sprouting phenomena [100,101]. Nanomaterials play another critical aspect [102]. For example, nanocarriers can be used to deliver the drug directly to the site of the lesion and thus avoiding the need for systemic administration with all the advantages that this entails: less toxicity, less expenditure of resources, longer-lasting release. Other elements, such as electrospun nanofibers, self-assembled nanofibers, and carbon nanotubes, can be used for structural purposes, creating scaffolds that can promote and accommodate the regeneration of neuronal tissue [103,104,105,106]. Finally, another encouraging technique used for the repair of damage to the nervous tissue consists of the use of fusogens, a heterogeneous group of chemical agents capable of promoting cell fusion [107]. They are classified into two large groups based on their mechanism of action. The first group induces fusion through cell aggregation, while the second through modifications of the electrical charges of the plasma membranes [108]. The most promising results are observed around peripheral nerve repair, where it seems to be able to continuously repair solutions at the level of the axonal membrane [109,110]. Among the different fusogens, the most used and promising seems to be polyethylene glycol (PEG), used in experiments for the repair of both peripheral nerves and in cases of spinal cord damage [107,111]. The direct application of PEG to the site of spinal cord damage can repair cell membranes, mitigate oxidative stress and the formation of the glial scar, and promote axonal regeneration, managing to restore motor function [112]. Furthermore, PEG cross-linking produces hydrogels that can act as delivery vehicles for growth factors and cells, such as bone marrow stromal cells that are able to modulate the inflammatory response and support neural tissue repair [113]. Despite the relative safety of PEG, some critical issues have been reported in the use of this biopolymer, such as the production of anti-PEG antibodies observed in animal and human models capable of stimulating an immune response [114,115]. Furthermore, only a modest therapeutic efficacy has been demonstrated for the use of fusogens during a series of chronic neurobehavioral experiments [116].

## 5. Conclusions

Spinal cord injury (SCI) results in direct and indirect damage to neural tissues, which in turn determines motor and sensory dysfunction, dystonia, and pathological reflex, ultimately leading to paraplegia or tetraplegia. After losing cells, axon regeneration failure, and time-sensitive pathophysiology make tissue repair difficult.

Stem cells-based therapy has some neuroregenerative and neuroprotective effects in SCI treatment. However, it presents some safety concerns. First, cell therapy-related immunotoxicity, immunogenicity, and tumorigenicity are often discussed in preclinical studies. Second, limited cell survival and limited integration were common obstacles in previous studies with different experimental designs, including cell number, timing of treatment, and strategies of transplantation. Third, it is essential to ensure the genetic stability, generation consistency, and storage safety of stem cells. Moreover, the mechanism of the effects and biological properties should be further investigated to guide the clinical application. Finally, small sample size, limited supervision, and poor quality are the common problems of most registered clinical trials that hinder the development of stem cell therapy. Standard protocols are difficult to confirm due to the heterogeneity of the injury type and level, the time of treatment, and the different number of transplanted cells.

Encouraging preclinical studies led to early clinical deployment, but the results were mixed. One specific type of stem cell achieves only a limited therapeutic effect. Therefore, many researchers are committed to enhancing the efficacy of stem cells. The use of genetic engineering technology, cell coupling, combinational therapy with neuroprotective agents, trophic factors, biomaterials, and rehabilitation may help improve the therapeutic effectiveness of stem cells in heterogeneous patient populations.

## Figures and Tables

**Figure 1 ijms-23-12996-f001:**
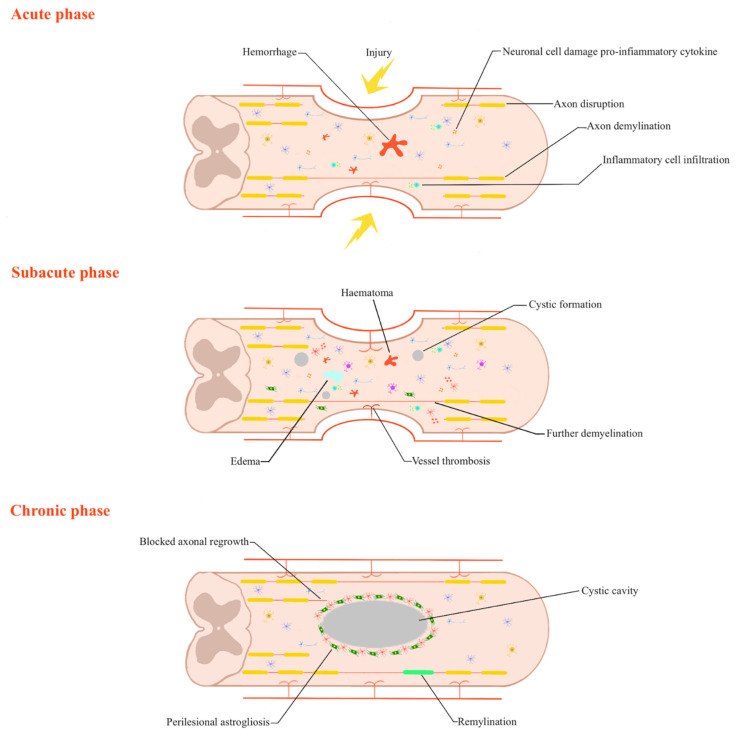
Phases of spinal cord injury. The acute phase occurs from 2 h to 48 h after injury and comprises increasing inflammation, edema, and hemorrhage. It is caused by free-radical generation, ionic dysregulation, excitotoxicity (owing to glutamate-mediated pathways), immune-related neurotoxicity. The subacute phase, which occurs from approximately day 2 to 2 weeks after injury, refers to the phagocytic response to clear cellular debris and initiate early axonal growth. During this phase, damaged astrocytes undergo cellular edema and necrosis, whereas astrocytes on the periphery of the injured tissue proliferate and function to reestablish ionic hemostasis and the blood-brain barrier and to restrict immune cell inflow. At this stage, the initial scar formation occurs. The chronic phase starts at 6 months after injury. It is characterized by further scar maturation and syrinxes formation.

**Figure 2 ijms-23-12996-f002:**
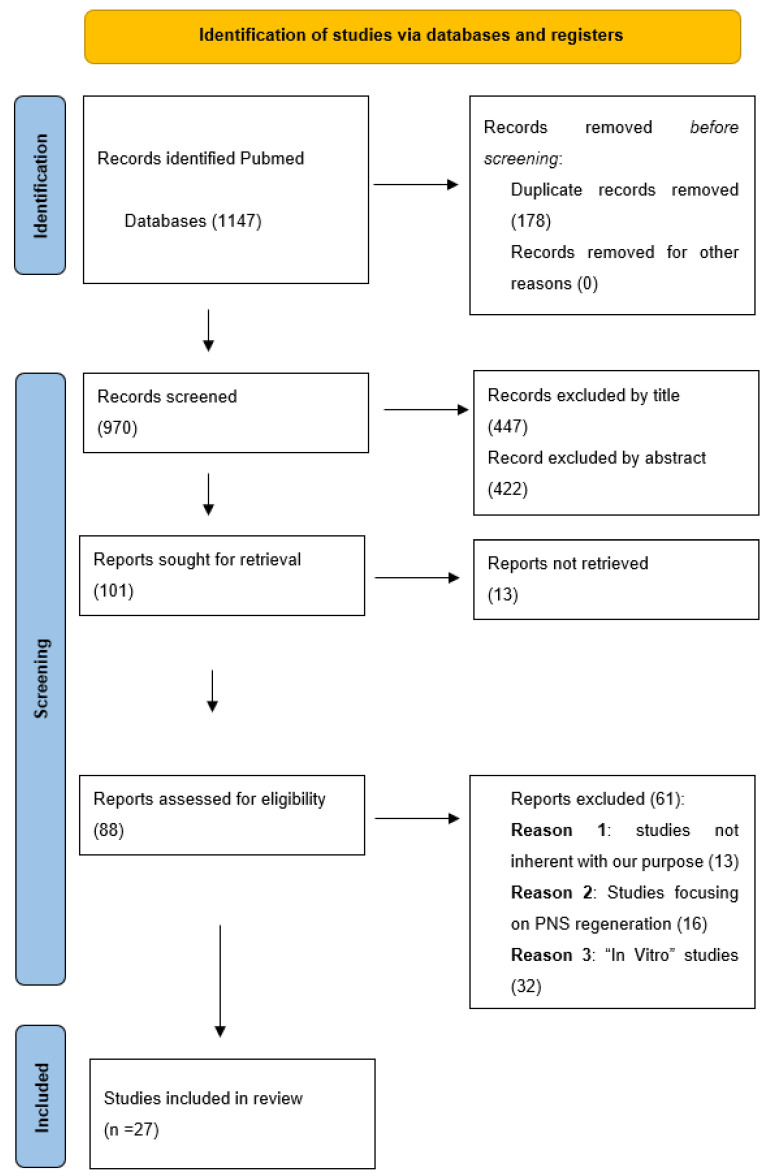
PRISMA flow-chart of included studies and selection process.

**Figure 3 ijms-23-12996-f003:**
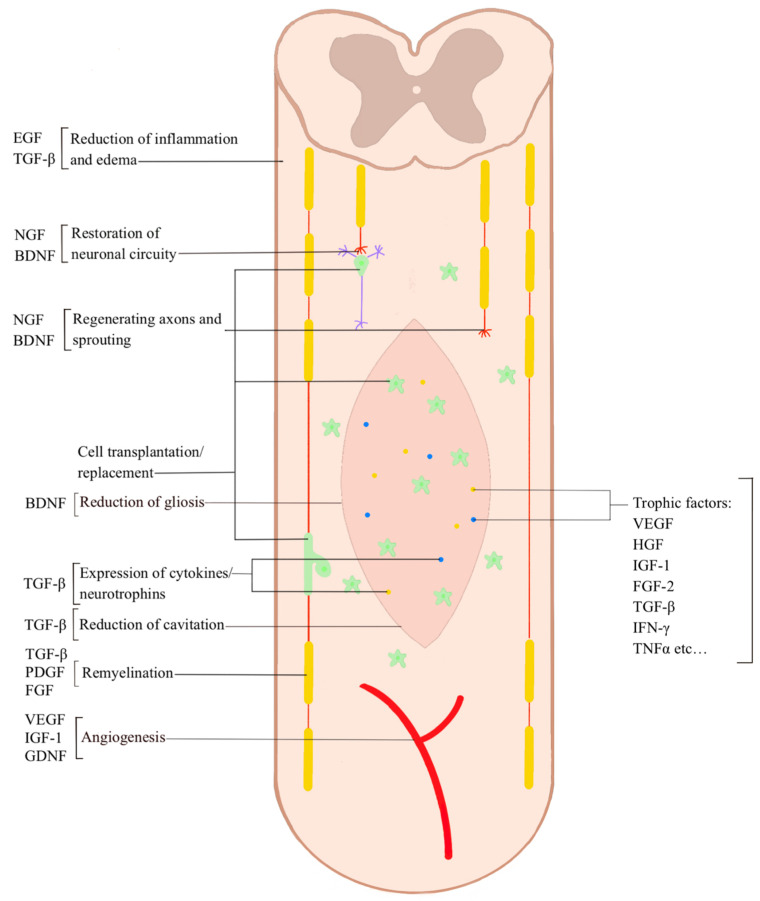
Stem cell mechanism of action. Stem cells enhance neuronal regeneration through cell replacement and the release of neurotrophic factors, limit scar formation and gliosis and inhibit the inflammatory microenvironment resulting from SCI through the release of anti-inflammatory cytokines.

**Table 1 ijms-23-12996-t001:** Main Results Obtained from Studies Included.

N°	Author	Year	Country	Study Design	Evidence Obtained
1	Ciciriello et al. [32]	2018	USA	Prospective experimental study on in vivo rats model	NSC from E14 mice on multichannel PLG scaffold increases density of axons and myelin regeneration and leads to a more rapid and functional recovery
2	Wan et al. [33]	2017	China	Prospective experimental study on in vivo rats model	NSC are cytocompatible with selfpolymerize dendritic polypeptide scaffold
3	Yang et al. [34]	2017	China	Prospective experimental study on in vivo rats model	Co-transplantation of ASCs and MSCs in a multichannel polymer scaffold leads to a better recovery after SCI
4	Führmann et al. [35]	2018	Canada	Prospective experimental study on in vivo rats model	chABC influence the injury environment such that neuronal differentiation or survival is favored. No functional repair was observed
5	Zhou et al. [36]	2018	China	Prospective experimental study on in vivo rats model	ASCs and (or) iPS-NSCs grow well on PCL scaffolds. Transplantation reduced the volume of the lesion cavity and improved the locomotor recovery of rats
6	Yan et al. [37]	2021	China	Prospective experimental study on in vivo rats model	Mbp supports axonal regeneration from mammalian NPCs through the novel 7Mbp/L1cam/Pparγ signaling pathway
7	Yuan et al. [38]	2021	China	Prospective experimental study on in vivo rats model	Cell-adaptable neurogenic (CaNeu) hydrogel as a delivery vehicle for ADSCs enhances axonal growth and leads to improved motor recovery in rats, also establishing an anti-inflammatory microenvironment
8	Hwang et al. [39]	2019	South Korea	Prospective experimental study on in vivo rats model	Glial cell line-derived neurotrophic factor (GDNF) augments the therapeutic effects of Neural stem/progenitor cells (NSPCs) in SCI
9	Farrag et al. [40]	2018	USA	Prospective experimental study on in vivo rats model	Encapsulated rat aNSCs in hydrogels implanted in the backs of rats in the cervical, thoracic, and lumbar region, expressed region-specific Hox genes corresponding to their region of implantation
10	Stewart et al. [41]	2017	USA	Prospective experimental study on in vivo rats model	Overexpression of SDF-1 by MSCs can enhance the migration of NSCs in vitro. Although only modest functional improvements were observed following transplantation of SDF-1-MSCs in vivo
11	Nori et al. [42]	2018	Canada	Prospective experimental study on in vivo rats model	Reprogrammed human NPCs biased toward an oligodendrogenic fate (oNPCs) in combination with sustained delivery of ChABC using an affinity release strategy in a cross-linked methylcellulose biomaterial leads to a better recovery in chronically injured spinal cords
12	Tian et al. [43]	2017	China	Prospective experimental study on in vivo rats model	Engineered nerve complex using acellular scaffolds to deliver placenta-derived stem cells (PMSCs) into the injury gap, showed enhanced regeneration, structurally and functionally.
13	Baklaushev et al. [44]	2019	Russia	Prospective experimental study on 2 exemplars of Macacus rhesus	A two-component matrix SPRPix, based on platelet-rich plasma (PRP) and an anisotropic complex scaffold of recombinant spidroins and polycaprolactone (rSS-PCL) induced a dramatically stimulated proliferation and neuronal differentiation of the drNPCs matrix in the NHP brain and spinal cord.
14	Babaloo et al. [45]	2019	Iran	Prospective experimental study on in vivo rats model	Animals implanted with PCL/gelatin scaffolds seeded with co-hEnSC demonstrated the most progressive recovery of hindlimb functions in comparison to the control group
15	Kourgiantaki et al. [46]	2020	Greece	Prospective experimental study on in vivo rats model	Grafts based on porous collagen-based scaffolds (PCSs), can deliver and protect embryonic NSCs at SCI sites, leading to significant improvement in locomotion recovery
16	Salarinia et al. [47]	2020	Iran	Prospective experimental study on in vivo rats model	Axon regeneration increased, cell apoptosis decreased and locomotor function improved when PRP and AD-MSCs were applied together, in comparison to when either AD-MSCs or PRP were used alone
17	Tsai et al. [48]	2018	Taiwan	Prospective experimental study on in vivo rats model	Systemic administration of conditioned medium from MSCs (MSCcm) induce a long-lasting neuroprotective effect on SCI rats and may provide an environment more conducive to corticospinal axonal regrowth after spinal cord injury
18	Fan et al. [49]	2018	China	Prospective experimental study on in vivo rats model	Gelatin methacrylate (GelMA) hydrogel with iPSC-derived NSCs (iNSCs) significantly promoted functional recovery
19	Zahir et al. [50]	2008	Canada	Prospective experimental study on in vivo rats model	NSPCs seeded in chitosan tubes survive well, differentiate, and allow axonal regeneration through the tubular construct in a severe, complete spinal cord transection injury model
20	Xia et al. [51]	2013	China	Prospective experimental study on in vivo rats model	Co-transplantation of NSCs with SCs seeded within a directional PLGA scaffold has a beneficial function in cell survival, differentiation, axonal regeneration and myelination, and motor function recovery. However, regenerated axons have a limited contribution to motor function recovery
21	Ribeiro-Samy et al. [52]	2013	Portugal	Prospective experimental study on in vivo rats model	PHB-HV scaffolds reveal their ability to support the culture of CNS-derived cells and mesenchymal-like stem cells from different sources, also showing they are well tolerated by the host tissue, and do not negatively impact left hindlimb locomotor function recovery
22	S. Wilems et al. [53]	2015	USA	Prospective experimental study on in vivo rats model	A multifactorial approach, with scaffolds containing pMNs, but not anti-inhibitory molecules, showed survival, differentiation into neuronal cell types, axonal extension in the transplant area, and the ability to integrate into host tissue. However, the combination of pMNs with sustained-delivery of anti-inhibitory molecules led to reduced cell survival and increased macrophage infiltration
23	H. All et al. [54]	2015	USA	Prospective experimental study on in vivo rats model	Transplanted IPS-derived OPs resulted in a significant increase in the number of myelinated axons in animals that received a transplantation 24 h after a moderate contusive spinal cord injury
24	Young Hong et al. [55]	2014	South Korea	Prospective experimental study on in vivo rats model	iNSCs transplantation effectively reduced the inflammatory response and apoptosis in the injured area. Furthermore, it also promoted the active regeneration of the endogenous recipient environment in the absence of tumor formation
25	Lee et al. [56]	2014	China	Prospective experimental study on in vivo rats model	Exogenous melatonin administration combined with physical exercise increases histological and behavioral recovery. Additionally, this dual treatment appears to increase nestin-positive eNSPCs, driving effective reconstructed neuronal differentiation
26	Lai et al. [57]	2014	China	Prospective experimental study on in vivo rats model	Transplantation of the GS scaffold promotes exogenous NSC-derived myelinating cells and SCs to form myelin in the injury/transplantation area of the spinal cord
27	Wang et al. [58]	2010	China	Prospective experimental study on in vivo rats model	Co-transplantation of NSCs and OECs might have a synergistic effect on promoting neural regeneration and improving the recovery of locomotive function

**Table 2 ijms-23-12996-t002:** Pros and Cons of Different Stem Cell type.

Stem Cell Typology	Advantages	Disadvantages
BM-MSCs	Secrete neurotrophic factors Promote axonal regeneration Reduce astroglial scarring density and inflammatory reaction Reduce BSCB leakage Regulate autophagy Alleviate neuropathic pain	The effects of individual cell transplantation are enhanced by co-transplantation with cells from other sources (SCs, OECs) Little therapeutic effect (timing of MSC transplantation)
U-MSCs	Readily available Inhibit glial scar and decrease reactive astrocytes Attenuate ischemic compromise of the spinal cord Improve muscle tension, bladder function, and urine control	Co-transplantation may complement and synergize to improve single-cell therapies (U-MSCs, hNSCs)
AD-MSCs	Protect neurons Promote cell survival and tissue repair Suppress immune activity and secrete anti-inflammatory factors Activate angiogenesis Reduce the formation of cavities	No site lesion reduction Lack of standard protocols for cell generation No clear cell characteristics No clear underlying mechanism
NSCs and NPCs	Increase neuroprotective cytokines and improve cell proliferation Increase myelination Modulate the inflammatory response Promote respiratory recovery	Modified NSCs may exhibit better therapeutic efficacy than naïve cells Functional recovery was limited
ESCs	Enable axons to pass CSPG Support nodal architecture Attenuate neuropathic pain	Undifferentiated form is rarely used due to the risk of tumorigenicity May result in tumor formation May be genetic changes during the cell culture process
iPSCs	Improve neurotrophic factor secretion Promote axonal sprouting and remyelination Promote synapse formation Inhibit glial scar formation Reduce lesion size	Different transplantation regions may lead to different effects (intraspinal implantation vs. intrathecal implantation) May result in tumor formation Limitations with graft survival or time to transplant Prohibitively high cost–benefit for developing treatments No standard protocols for collecting cells, for safe and effective routes of administration in clinical treatment

Abbreviations: BM-MSCs = Bone Marrow–Mesenchimal Stem Cells; OECs = olfactory ensheathing cells; SCs = Schwann Cells; BSCB = blood-spinal cord barrier; CSPG = Chondroitin sulfate proteoglycans; U-MSCs = Umbilical Mesenchimal Stem Cells; hNSCs = human Neural Stem Cells; NPCs = Neural Progenitor Cells; AD-MSCs = Adipose-derived Mesenchimal Stem Cells; ESCs = Embryonic Stem Cells; iPSCs = Induced Pluripotent Stem Cells.

## Data Availability

Not applicable.

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
