# Peer review of "Stem Cell Strategies in Promoting Neuronal Regeneration after Spinal Cord Injury: A Systematic Review"

_ijms, 2022, doi:10.3390/ijms232112996_

Round 1

Reviewer 1 Report

The review by Bonosi et al. provides an overview of stem cell therapy in spinal cord injury. The review is interesting and covers the last five years research on different stem cell types for spinal cord injury therapy in animal models.

1. For spinal cord injury research, 5 years is relatively narrow time span, especially for clinical study. It would be helpful to extend the time range to 10 year for clinical studies, and to add the corresponding clinical findings to table 2.

Author Response

Dear Reviewer,

I hope this email finds you well

First of all, I would like to thank you for taking the time to read our work

In agreement with your advice, we decided to extend the time inclusion criterion, considering the literature on this subject produced since 2008.

Regarding the evidence obtained from each study, these have been included and summarized in the last column of Table 1.

Best regards

Reviewer 2 Report

This is a reasonable review of the recent literature addressing the use of various types of stem cells in treating spinal cord injury.  As such, it serves as a common source and summary of relevant references for those working in the field who do not have the time or inclination to read the primary literature.  There are a few typographical and grammatical issues that need to be addressed (the reference to "type of astrocytes" on line 68 appears to imply that Schwann cells are a type of astrocyte) and many of the citations have incomplete information in the reference list, e.g., refs 10, 56, 59, 60, 61, and others.  The main limitation of the review is that there is little in the way of critical evaluation of the primary reports although the framework for the need for future work in the field is provided.

Author Response

Dear Reviewer ,

I hope you are doing well

We have corrected the typo in line 68; in fact, Schwann cells do not represent a subtype of astrocytes, but rather the subpopulation of glial cells most represented at the PNS level.

We then improved all references with the missing information.

In agreement with his suggestions we also implemented the review by considering papers pertaining to this focus published since 2008, to give a broader view of what progress has been made in this fascinating research field. This is a review on preclinical phase studies conducted in animal models, and therefore the evidence obtained could potentially be discordant in human application. Certainly, stem cell therapy has some interesting advantages that make their application in the clinical setting worthy of further investigation, in addition to the exciting possibility of repairing nerve tissue. Our first aim is to give an overall view on the pros and cons of this therapeutic strategy, analyzing the evidence obtained from the studies conducted so far (summarized in Table 1 and deepened in the various sections of the discussion).

Kind regards

Reviewer 3 Report

The presented manuscript attempts to provide an up-to-date summary of current researches, challenges, and future directions for stem cell therapy in spinal cord injury (SCI) models, providing an overview of this continually evolving and promising field.

1. The paper does not disclose the issue of unsolved problems of SCI repair as well as the purposes of cell therapy. For example, how is cell therapy better than fusogens administration?

2. The lack of graphics does not allow a quick dive into the topic. Figures are needed to illustrate the requirements for SCI repair.

3. The 18 studies included in the study are too few to title the review as systematic. The authors need to address this problem. For example, authors may conduct a more in-depth analysis of the papers and identify previously unknown findings.

The manuscript as presented cannot be recommended for publication.

Author Response

Dear Reviewer 3,

I hope you are doing well

I would really thank you for taking the time to read our work and I have appreciated your advice to improve our work. 

The paper does not disclose the issue of unsolved problems of SCI repair as well as the purposes of cell therapy. For example, how is cell therapy better than fusogens administration? 

Reply 1: We have implemented the discussion by adding a section regarding fusogens therapy, its advantages and disadvantages, and its application, to give a more wide-ranging view regarding possible nerve tissue regeneration techniques. Regarding the issue of SCI repair this appears to be still unresolved, and stem cell therapy represents, in this sense, a first step toward a possible solution, although it has some limitations that are highlighted in the final paragraph of the discussion.

The lack of graphics does not allow a quick dive into the topic. Figures are needed to illustrate the requirements for SCI repair.

Reply 2: We have added two pictures (Figure 1 and 3 in the main text). The first is about the phases of spinal cord injury and damage mechanisms, while the second depicts the mechanisms of action of stem cell therapy to restore neural anatomy and function.

The 18 studies included in the study are too few to title the review as systematic. The authors need to address this problem. For example, authors may conduct a more in-depth analysis of the papers and identify previously unknown findings.

Reply 3: in agreement with his/her criticisms we have implemented the review by considering papers pertaining to this focus published since 2008, to give a broader view of what progress has been made in this research field.

Sincerely

Round 2

Reviewer 1 Report

No further questions.

Author Response

Dear Reviewer,

I hope this email finds you well

Thank you very much for your suggestions and your time

Kind regards

Reviewer 3 Report

The article has become a bit improved, but this is not enough. The fact is that the problem of spinal cord repair is extraordinarily important, so superficial or poor papers in this field are completely unacceptable.

1. Figs. 1 and 3: These figures do not reflect the mechanisms of spinal cord recovery after injury. The figures need to be revised.

2. Lines 181-183: The authors should disclose the full query formula. This is necessary for readers and reviewers to assess the adequacy of the search. The results of the search should be attached as an Excel file in Supplementary.

3. Line 266: "Duplicate records removed" - if authors used a single query, the duplicate records should be absence.

4. According to this paper, microglia are not involved in spinal cord repair.

5. Section "4.5. Alternative methods to promote repair and regeneration of nerve tissue" is not enough to uncover the alternative methods. There is very little discussion.

Author Response

Dear reviewer,

I hope this email finds you well

Thank you very much for your comments.

We have revised the manuscript according to your suggestions

Query 1: “Figs. 1 and 3: These figures do not reflect the mechanisms of spinal cord recovery after injury. The figures need to be revised”.

Reply 1: In agreement with your suggestions, we have revised the pictures to make the images and their purpose within the text more detailed and understandable. The aim of Fig. 1 is to schematize the various stages of spinal cord injury and the main cellular components involved rather than recovery phases; Fig. 3, on the other hand, aims to exemplify the action of stem cells in SCI by naming the various mechanisms by which they act and some of the biomolecules involved. Both processes are then further clarified and better defined within the main text. 

Query 2: “Lines 181-183: The authors should disclose the full query formula. This is necessary for readers and reviewers to assess the adequacy of the search. The results of the search should be attached as an Excel file in Supplementary.”

Reply 2: Following your criticisms, we have attached in the supplementary materials the search strings used initially with the results obtained for each of them. In the uploaded excel file you can find all screened articles after removing duplicates.

Query 3: “Line 266: "Duplicate records removed" - if authors used a single query, the duplicate records should be absence.”

Reply 3: We did not use a single search string but five different search strings. The results obtained for each string were exported to an excel file, and using the "remove duplicates" function, we received the total number of items to be evaluated for review.

Query 4: “According to this paper, microglia are not involved in spinal cord repair”.

Reply 4: In accordance with your suggestions, we have enriched the introduction (line 89 - 124) by better specifying the role of microglia and immune cells in the process of spinal cord injury and repair. We agree with the reviewer about the fundamental role of this cellular component; so, we decided to mention it but not to elaborate on it further as it is not in line with the objective of our paper, which is to give an insight into stem cell therapy in spinal cord injury at the preclinical level.

Query 5:” Section "4.5. Alternative methods to promote repair and regeneration of nerve tissue" is not enough to uncover the alternative methods. There is very little discussion.”

Reply 5: We understand the reviewer's criticisms and because of this we have implemented the final paragraph of the discussion regarding alternative strategies to promote neuronal regeneration (lines 440 - 457). Again, we have chosen not to deepen this broad topic to avoid straying from our primary objective.

We hope to have addressed the reviewer’s criticism

Sincerely